# Dietary Fiber, Whole Grains, and Head and Neck Cancer Prognosis: Findings from a Prospective Cohort Study

**DOI:** 10.3390/nu11102304

**Published:** 2019-09-27

**Authors:** Christian A. Maino Vieytes, Alison M. Mondul, Zonggui Li, Katie R. Zarins, Gregory T. Wolf, Laura S. Rozek, Anna E. Arthur

**Affiliations:** 1Division of Nutritional Sciences, University of Illinois at Urbana-Champaign, Urbana, IL 61801, USA; 2Department of Epidemiology, University of Michigan, Ann Arbor, MI 48103, USA; 3Department of Food Science and Human Nutrition, University of Illinois at Urbana-Champaign, Urbana, IL 61801, USA; 4Department of Environmental Health Sciences, University of Michigan, Ann Arbor, MI 48103, USA; 5Department of Otolaryngology, University of Michigan, Ann Arbor, MI 48103, USA; 6Carle Cancer Center, Carle Foundation Hospital, Urbana, IL 61801, USA

**Keywords:** epidemiology, survivorship, obesity, cancer, fruit, vegetables, diet, nutritional epidemiology, cancer survivors

## Abstract

No studies, to date, have examined the relationship between dietary fiber and recurrence or survival after head and neck cancer diagnosis. The aim of this study was to determine whether pretreatment intake of dietary fiber or whole grains predicted recurrence and survival outcomes in newly diagnosed head and neck cancer (HNC) patients. This was a prospective cohort study of 463 participants baring a new head and neck cancer diagnosis who were recruited into the study prior to the initiation of any cancer therapy. Baseline (pre-treatment) dietary and clinical data were measured upon entry into the study cohort. Clinical outcomes were ascertained at annual medical reviews. Cox proportional hazard models were fit to examine the relationships between dietary fiber and whole grain intakes with recurrence and survival. There were 112 recurrence events, 121 deaths, and 77 cancer-related deaths during the study period. Pretreatment dietary fiber intake was inversely associated with risk of all-cause mortality (hazard ratio (HR): 0.37, 95% confidence interval (CI): 0.14–0.95, *p*_trend_ = 0.04). No statistically significant associations between whole grains and prognostic outcomes were found. We conclude that higher dietary fiber intake, prior to the initiation of treatment, may prolong survival time, in those with a new HNC diagnosis.

## 1. Introduction

Head and neck squamous cell carcinomas (HNSCCs) encompass malignancies of the upper aerodigestive tract epithelium, including squamous cell carcinomas of the oral cavity, oropharynx, hypopharynx and larynx [1]. Collectively, HNSCCs account for the 8th leading cause of cancer-related deaths in the United States [2]. The advancement of diagnostic technologies and treatment modalities, alone, have been unsuccessful in improving the 5-year survival rate for this class of tumors, with the estimated figure hovering at approximately 65% survival [2,3]. Despite the urgent need to clarify potential modifiable lifestyle factors that may influence HNSCC prognosis, investigative efforts in this regard have been and remain scant.

Dietary fiber consumption has received a notable degree of attention and is frequently cited as a lifestyle factor influencing cancer, although the bulk of evidence has come from examinations of colorectal cancer [4]. Fiber is believed to exert its effects through numerous mechanistic frameworks that may be contingent on the type of fiber consumed [4]. Within the context of colorectal cancer, these mechanisms may be succinctly summarized by the enhanced excretion of fecal carcinogens, modulation of the gastrointestinal microbiota and production of short-chain fatty acids (SCFAs), subsequent alterations to bile acid circulation and recycling, and others [4,5]. Systemically and in consideration of other cancer types, plausible mechanisms include the suppression of the insulin/insulin-like growth factor-1 (IGF-1) signaling cascade, effects on growth-promoting androgenic and estrogenic factors present in circulation, as well as the suppression of circulating proinflammatory cytokines [4,6,7,8]. Nevertheless, dietary fiber and its systemic mechanistic sequelae are multifaceted and their associations with measurable cancer phenotypes are likely to contribute in a synergistic manner [9].

The literature highlighting relationships between the consumption of dietary fiber and HNSCCs has been restricted to only a few, primarily case-control, examinations of HNSCC risk [10,11,12]. The sole longitudinal study to date that examined dietary fiber and risk of developing HNSCC reported an inverse association within the National Institutes of Health-American Association of Retired Persons (NIH-AARP) Diet and Health study cohort [13]. To our knowledge, no studies have yet examined associations between dietary fiber consumption and HNSCC prognosis. We previously identified and reported on associations between total carbohydrate intake and HNSCC prognosis in the University of Michigan Head and Neck Specialized Program of Research Excellence (UM HN-SPORE) longitudinal cohort [14]. While a higher consumption of total carbohydrate, glycemic load, or simple carbohydrates was associated with increased risks of all-cause and cancer-related deaths, notable among the findings was an inverse association between the consumption of starchy carbohydrate foods and risk of all-cause and HNSCC-specific mortality [14]. Dietary fiber is a non-caloric nutrient endemic to many of the elements defining the starchy carbohydrates category (potatoes, legumes, and other vegetables combined). Thus, the principal aim of this analysis was to examine pretreatment associations between total dietary fiber and whole grain consumptions and risk of HNSCC recurrence and mortality using data previously collected from the UM HN-SPORE longitudinal cohort. The study hypotheses were that the consumption of the aforementioned dietary factors would be inversely associated with HNSCC recurrence and mortality.

## 2. Materials and Methods 

### 2.1. Study Design and Recruitment

The University of Michigan Head and Neck Specialized Program of Research Excellence is a prospective longitudinal cohort study of newly-diagnosed head and neck cancer patients, recruited prior to the initiation of any treatment modality. Individuals in this cohort were diagnosed with tumors in the oral cavity, oropharynx, hypopharynx, or larynx. Participants were recruited for the purpose of collecting data on demographic, clinical and lifestyle variables in order to assess their associations with pertinent cancer-related outcomes. Recruitment took place between November 2008 through October 2014 and newly diagnosed, previously untreated HNC patients seen within the UM Hospital System were approached and solicited for inclusion into the study. Exclusion criteria included: (i) being less than 18 years of age, (ii) pregnant, (iii) non-English speaker, (iv) a diagnosis suggestive of mental instability, (v) diagnosis with another tumor of the non-upper aerodigestive tract, and (vi) diagnosis with another primary HNC during the last 5 years. Baseline and annual medical record reviews were conducted for surveillance of clinical factors including tumor site, cancer stage, treatment history, comorbidities, survival, and recurrence status. Baseline (pretreatment) measures included dietary records as well as data on health behaviors and epidemiological variables collected via a survey upon entry into the cohort. All study activities were approved and completed in accordance with standards approved by the Institutional Review Board of the University of Michigan Medical School and complied with the Helsinki Declaration of 1975. The IRB approval number for UM HN-SPORE, for which written consent was obtained to acquire and analyze the data, is HUM00042189.

### 2.2. Measurements

#### 2.2.1. Predictors: Dietary Intake of Fiber and Whole Grains

Baseline data obtained prior to the initiation of treatment for HNSCC included dietary records measured via the previously validated and self-administered 2007 Harvard adult food frequency questionnaire (FFQ) [15,16]. This 131-item FFQ is designed to estimate the usual intake of foods, beverages, and nutrients over the past year. All baseline nutrient intake values were determined according to the Harvard nutrient database, which draws from a number of nutrient databases for computing those figures [17]. 

Dietary fiber was defined as the total fiber content stemming from the consumption of all fruits, vegetables, cereal, and legume sources and was estimated according to the gravimetric method established by the Association of Established Agricultural Chemists (AOAC) [18]. Whole grain content was defined as that from grain-containing foods (breakfast cereals, rice, bread, and pasta) and encompassed intakes of intact bran, germ, and endosperm (and also took into consideration added bran and added wheat germ), as has previously been described [19]. Dietary fiber and whole grain intakes were each categorized into quintiles. The categorization of dietary variables considered their distributions in addition to the treatment of these variables in previous analyses [14,20]. All nutrient variables included in the analyses were energy-adjusted using the residual method described by Willet et al. [21]. Influential outliers for total caloric intake, dietary fiber and whole grain intakes were assessed using the Rosner method [22]. This analysis excluded participants reporting daily caloric intakes of <200 kcal (*n* = 11) or >5000 kcal (*n* = 8). The same method was applied to dietary fiber and whole grain variables, which ultimately resulted in the exclusion of individuals reporting intake values >51 g (*n* = 6) and >118 g (*n* = 6) for fiber and whole grains, respectively. A decision was made, a priori, to retain lower bound outliers for dietary fiber given that near-zero intakes are potentially feasible in this patient population given the ubiquity of nutrition impact symptoms and the ramifications they pose for obtaining the adequate consumption of key nutrients [23]. Individuals with full pages missing (*n* = 17) from any administered FFQ or having greater than 70 blank responses (*n* = 1) were omitted from the final sample. The final sample size for this analysis retained 463 study participants.

#### 2.2.2. Covariates

Covariate selection was based on prior knowledge of factors known to influence the outcomes of interest (recurrence and survival). Demographic control variables included age and sex. Clinically-relevant covariates included tumor stage (ranging from 0–IV), human papillomavirus (HPV) status (categorized as positive, negative, or unknown), and disease site (involving the larynx, oral cavity, hypopharynx, or oropharynx). Treatment modality had been given a priori consideration but was omitted to avoid collinearity given significant correlation with tumor stage (Spearman *r* = 0.37, *p* < 0.001). Lifestyle covariates included Body Mass Index (BMI) (kg/m^2^) (considered as a continuous variable), marital status (categorized as married, widowed, separated/divorced, or never married), smoking status (categorized as current/former and never), drinking status (categorized as never, former, or never), and highest level of education attained, (dichotomized as high school or less and some college or greater). Dietary covariates included total fat intake (categorized by tertiles) and glycemic load (categorized by quartiles). Fruit and vegetable intake was computed for each participant using an average of the following nutrient variables (servings/day): Total fruit, cruciferous vegetables, green/leafy vegetables, tomatoes, potatoes, legumes, dark/yellow vegetables, and “other” vegetables. Intake levels were subsequently stratified into quartiles. 

#### 2.2.3. Outcomes: Recurrence and Mortality

The primary outcomes, recurrence and mortality, are time-to-event in nature and were assessed using available data on time-to-recurrence, survival time, recurrence status, and death status. These data were collected at the annual medical chart reviews and death status was adjudicated through one or a combination of the following sources: The Social Security Death Index and LexisNexis, updates to medical and survey data at each of the follow-up time-points, as well as through notification from family, other physicians, or medical record reviews. Cause of death was recorded in instances where obtaining these data were feasible. Recurrence/persistence-free time and survival time used the date of diagnosis as the initiation of the follow-up period. Survival time was censored at February 1, 2014 and recurrence-free time was censored according to the date of the last known medical record review for a given participant. Loss to follow-up was rectified by censoring participants to the date of their last reported status.

### 2.3. Statistical Analysis

Descriptive statistics were tabulated for demographic, clinical, and behavioral characteristics. Spearman correlation coefficients were used to assess collinearity amongst the chosen covariates. Survival functions were initially modeled using Kaplan–Meier survival functions. The Log-rank test was used to evaluate for significant differences between two or more survival curves, each corresponding to the different quintiles of fiber or whole grains. 

Risks of tumor recurrence and all-cause or cancer-specific mortalities were assessed using Cox proportional hazard models, employed to estimate hazard ratios (HRs) for each quintile of dietary fiber or whole grain intakes and their corresponding 95% confidence intervals (CIs). The proportional hazards assumption was tested by assessment of Schoenfeld residuals and subsequently fitting models with time-dependent covariates and evaluating for the presence of any significant interactions. No statistically significant violations to the model assumptions were identified.

Three models were constructed. Model 1 was defined as the basic model and included age and sex as covariates. Model 2, described as the clinicopathological model, further adjusted for clinically-relevant factors: HPV status, tumor stage, and site. Model 3, the fully adjusted model, added lifestyle and dietary covariates, including BMI, smoking status, drinking status, total fat intake, glycemic load, and level of education attained. The first quintile of dietary fiber or whole grains was set as the referent. Tests for linear trend were conducted by assigning the median value of the respective quintile to each participant and modeling as a continuous variable. Propensity score weights were computed to improve covariate balance and to further evaluate any significant associations that arose from non-weighted models. These weights were computed using the inverse probability of treatment weights (IPTW) and incorporated into the outcome models as previously described [24,25,26].

To assess for interactions between the predictors and covariates on recurrence and all-cause or cancer-specific mortality risk, fully adjusted models were stratified by sex, stage, disease site, BMI, and smoking status. In addition, the presence of interactions was assessed with interaction terms included in the model. The significance of these interaction terms was tested using the Likelihood Ratio Test.

Non-linearity and dose-dependence were assessed with restricted cubic spline analyses of hazard ratios across fiber or whole grain intakes using three knots set at the medians of the first, third and fifth quintiles (using the SAS LGTPHCURV9 Macro) [27]. The minimum intake values for fiber and whole grains were set as referents. As a further measure of sensitivity and to address the potential bias introduced by the degree of cases with an “unknown” HPV classification, mode imputation was conducted whereby unknown cases were allocated to the most prevalent category. 

All statistical tests were two-sided and conducted at α = 0.05. In order to correct the experiment-wise Type I Error probability for the multiple comparisons carried out in the stratified subanalyses, the Holm–Bonferonni method was employed to adjust the level of α. Statistics and analyses were generated and performed using the Statistical Analyses System (SAS Institute, Cary, NC, USA).

## 3. Results

### 3.1. Participant Characteristics

Epidemiological characteristics of the cohort are summarized in Table 1. The vast majority of participants were males (74.6%), identified as non-Hispanic white (94.8%), and reported an education level of some college or beyond (65.2%). The average age at HNSCC diagnosis was 61 years. The most prevalent disease site affected was the oropharynx (39.8%) and being a current/former smoker (71.4%) or a current/former drinker (92.4%) were both frequently reported behaviors.

Participant characteristics according to the quintile of dietary fiber or whole grain intakes are provided in Table 2. Individuals in the fifth quintile for dietary fiber consumption tended to be more educated, smoke and drink less, and were also less likely to have a tumor diagnosed at an advanced stage when compared to those with lower intakes. BMI and total caloric intake did not appear to vary substantially across the quintiles although higher fiber intake trended positively with both fruit/vegetable consumption and glycemic load. There were larger proportions of females in the higher quintiles of fiber consumption. The distributions of participant characteristics were similar for whole grains, with the exception of sex. Moreover, participants with stage III/IV tumors were more likely to have the highest consumption of whole grains compared to the lowest.

### 3.2. All-Cause and Cancer-Specific Mortality

The follow-up period culminated in 1499.03 person-years of data with 112 recurrence events, 121 deaths documented from all-causes, 77 cancer-related deaths, and a median survival time of 3 years. Implementing listwise deletion excluded seven and nine participants in the fully adjusted models for fiber and whole grains, respectively. Kaplan–Meier curves for the associations between dietary fiber or whole grain intakes and recurrence or survival are displayed in Figure 1. The inspection of survival curves stratified by quintiles revealed visually discernable differences between the 3–5th quintiles and 1–2nd quintiles. Consequently, a decision was made, a posteriori, to collapse each of those strata into binary categories and rerun the analysis for both fiber and whole grains, which showed a significant difference between the two fiber groups (*p*_Log-rank_ = 0.03).

After adjustment for age and sex (i.e., the basic model), a higher daily intake of total fiber was significantly and inversely associated with risk of all-cause mortality (Table 3). The results were unchanged with further adjustment in the clinicopathological and fully adjusted models (Table 3). Tests for linear trend were also significant in each of these models and, thereby, suggestive of a linear relationship between dietary fiber intake and risk of all-cause mortality (*p*_trend_, fully adjusted = 0.04). These findings were further corroborated by a null finding from the restricted cubic splines analysis (*p*_non-linear_ = 0.17, Appendix A). The magnitude of the association was similar when we examined cancer-specific mortality, although these findings were not statistically significant. These HR estimates were also consistent and changed negligibly with further multivariable adjustment. Propensity score weighting for dietary fiber bolstered the magnitude and significance of the reported associations for fiber and all-cause mortality, including that of the linear trend that was tested (Table 4). Mode imputation of HPV status did not alter the estimates for any of the models and results that are described (Appendix A). When we examined dietary whole grain intake, we observed a suggestion of an inverse association for both all-cause and cancer-specific mortality, although these findings were not statistically significant (Table 3).

### 3.3. Recurrence

The relationships between dietary fiber and whole grains and recurrence were nonsignificant. The results from the analysis with fiber and whole grains are highlighted in Table 5. Multivariable models resulted in parameter estimates indicating non-significant protective associations between the higher consumption of dietary fiber or whole grains and risk of recurrent disease when comparing the 5th and 1st quintiles of intake.

### 3.4. Subgroup Analyses

Interactions between fiber and whole grain intakes and clinical variables were assessed in stratified analyses. The results are found in Appendix A. The significance of the interaction terms, comparing nested models, are summarized in Appendix A. There were no significant interactions for any of the factors with any outcome examined.

## 4. Discussion

In the present analysis, we found that HNSCC patients with the highest pretreatment fiber intake had a 63% decreased risk of all-cause mortality relative to those with the lowest consumption. These findings did not appear to be confounded by other factors examined. A propensity score-weighted model supported these findings and demonstrated a strengthened association as compared to the non-weighted model. A similar, albeit non-statistically significant association was observed for cancer-specific mortality. These associations appeared to follow a linear, dose-dependent trend. Analyses examining pretreatment whole grains suggested non-significant decreases in all-cause mortality, cancer-specific mortality, and recurrence with higher whole grain intake. 

Similar to the results we report, an analysis of fiber intake in pooled case-control data from the International Head and Neck Cancer Epidemiology Consortium (INHANCE) revealed an inverse association with HNSCC incidence within all subsites examined [10]. Lam et al. reported an inverse association of HNSCC risk with increasing fiber intake in the NIH-AARP Diet and Health Study cohort [13]. A subanalysis revealed a significant reduction in HNSCC risk for women, although the risk estimates were attenuated and non-significant for men, in contrast to our findings, which did not ascertain a fiber by sex interaction. In a previous analysis conducted by our research team, it was found that the consumption of total carbohydrates was positively associated with risk of all-cause mortality [14]. Additionally, the higher consumption of simple carbohydrates, glycemic load, or simple sugars were each associated with an increased risk of all-cause mortality. Total carbohydrates and simple sugars were associated with elevated risk of HNSCC-specific mortality. Here, we further delineate the broader classification of dietary carbohydrates by considering fiber and whole grains and their associations with HNSCC recurrence and survival. This examination adds to the growing body of evidence highlighting the importance of considering subclasses of dietary carbohydrates individually and to the literature supporting a beneficial role of fiber for cancer risk and survival. Furthermore, the analysis conducted herein adds to the paucity of such studies conducted specifically after a cancer diagnosis and in those with HNSCC, a critically underserved cancer population. 

Scientific understanding of the systemic effects of dietary fiber on cancer remains limited. The scope of evidence suggests that systemic effects are likely arbitrated through modulation of the gastrointestinal microbiota as a result of prebiotic and fermentable qualities of particular fiber types [5]. A probable and subsequent consequence of this is a generalized decrease in systemic inflammation attributable to the effects of SCFA. Concentrations of SCFA have previously been measured and described in the circulatory system, and experimental evidence has highlighted the influence they have on gene expression [28,29,30]. These effects may extend and explain the associations we report here with HNSCC, albeit there is no evidence in the literature, as yet, to validate this hypothesis. To date, animal models comprise the foundation of our understanding of these relationships, and the generalizability of these results to humans is limited [31,32]. Another plausible and implicated pathway is the insulin/IGF-1 axis. The relevance of targeting this pathway for HNSCC prognosis has previously been contemplated [33]. Observational data has shown that higher prediagnostic serum IGF-1 predicted progression and a higher rate of developing a second primary tumor in head and neck cancer patients [34]. The consumption of prebiotics and dietary fibers have been associated with a decrease in markers of hyperinsulinemia and insulin resistance as well as with a favorable increase in IGF Binding Protein-3 (IGFBP-3), the latter of which is understood to quench bioavailable IGF-1 in the serum, thus mitigating tumor progression [6,7,35,36,37,38].

The strengths of the present study include the longitudinal design, consideration of both all-cause and cancer-specific mortality, and adjustment for multiple variables known to influence the outcomes of interest, including tumor HPV-status [39,40,41]. Generally, higher fiber intake is often an indicator of a high-quality diet, rich in fruits and vegetables, which may potentially confound results. Thus, adjusting for fruit and vegetable intake in the present study minimizes the likelihood of such a bias having influenced the outcomes reported. Additionally, the UM HN-SPORE represents, to our knowledge, the largest and only longitudinal cohort study of HNSCC patients with comprehensive data collected on diet. 

Some limitations of this study are worth noting. First, the present analysis utilized pretreatment values for the consumption of dietary fiber and whole grains, which fails to take into account the dynamic nature of dietary patterns over time. It is possible that fiber consumption may decrease during and after cancer treatment in many patients as a result of disease- and treatment-related symptoms that may negatively affect the ability or desire to consume fibrous foods [23,42]. Second, our dataset lacked information on food sources of fiber (e.g., cereal-, fruit-, or vegetable-derived). Third and lastly, despite the multivariate analysis we conducted, residual confounding and reverse causation cannot be ruled out and the FFQ utilized for dietary assessment is susceptible to measurement error and systematic biases [43]. 

Fiber intake was, overall, markedly low for most participants in the study (median intake 17.4 g/day) compared to the 22.4 and 28g of daily fiber for women and men, respectively, recommended by the 2015–2020 Dietary Guidelines for Americans. This may have led to an attenuated estimate of the true association. Nevertheless, the median intakes we report are comparable to fiber intake levels across the nation in adults greater than twenty years of age, as reported from the National Health and Nutrition Examination Survey (NHANES) data [44,45].

## 5. Conclusions

In sum, this is the first prospective cohort study examining the relationships between the consumption of dietary fiber and whole grains with recurrence and survival in HNSCC. We conclude that a higher pretreatment consumption of dietary fiber may impart benefit for curtailing all-cause mortality in newly diagnosed HNSCC patients. These results can inform the development of randomized controlled trials examining dietary fiber interventions in HNSCC patients as well as subsequent efforts that focus on delineating how fiber affects prognosis and the mechanisms involved. More generally, these results add to the growing body of evidence that supports dietary fiber consumption as a modifiable lifestyle factor that may favorably influence cancer development and prognosis. 

## Figures and Tables

**Figure 1 nutrients-11-02304-f001:**
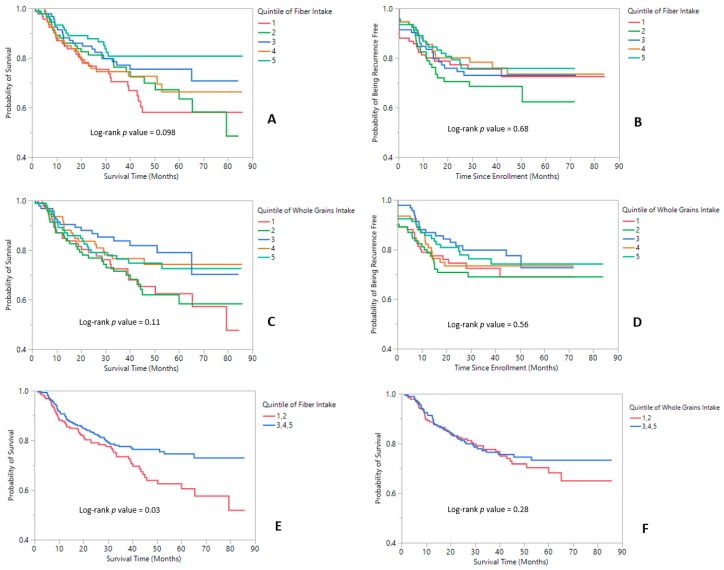
Kaplan–Meier Survival function plots for survival and recurrence. (**A**) Survival according to the quintile of fiber intake. (**B**) Recurrence according to fiber intake. (**C**) Survival according to whole grain intake. (**D**) Recurrence according to whole grain intake. (**E**) Survival according to a binary predictor (collapsing quintiles 1 and 2 or 3, 4, and 5 into separate categories) of fiber intake. (**F**) Survival according to a binary predictor (collapsing quintiles 1 and 2 or 3, 4, and 5 into separate categories) of whole grain intake.

**Table 1 nutrients-11-02304-t001:** Demographic, clinical, and behavioral characteristics of the study participants (*n* = 463).

Characteristic	Survivors # (%)
Age (year)	
Mean ± SD	61.1 ± 11.3
Min/Max	25/95
Sex ^b^	
Male	344 (74.6)
Female	117 (25.4)
Education ^c^	
High school or less	160 (34.8)
Some college or more	300 (65.2)
Race ^d^	
Non-Hispanic white	434 (94.8)
Other	24 (5.2)
Body Mass Index (BMI) (kg/m^2^)	
Underweight (<18.5)	20 (4.1)
Normal weight (18.5–24.9)	140 (30.2)
Overweight (25–29.9)	177 (38.2)
Obese (30+)	126 (27.2)
Site ^a^	
Oral cavity	173 (37.5)
Oropharynx	184 (39.8)
Hypopharynx	11 (2.4)
Larynx	94 (20.4)
Stage	
0, I, II	145 (31.3)
III, IV	318 (68.7)
HPV status ^a^	
HPV-negative	150 (32.5)
HPV-positive	73 (15.8)
Unknown	239 (51.7)
Treatment ^a^	
Surgery only	116 (25.1)
Radiation only	34 (7.4)
Surgery + adjuvant radiation or chemo	83 (18.0)
Chemotherapy + radiation	186 (40.3)
Chemotherapy only	14 (3.0)
Palliative or unknown	29 (6.3)
Smoking Status ^a^	
Current	168 (36.3)
Former	162 (35.1)
Never	132 (28.6)
Drinking status ^a^	
Current	319 (69.1)
Former	108 (23.4)
Never	35 (7.6)

^a^*n* = one missing, ^b^
*n* = two missing, ^c^
*n* = three missing, and ^d^
*n* = five missing.

**Table 2 nutrients-11-02304-t002:** Select epidemiological characteristics according to quintile (Q) of dietary fiber or whole grain intake.

**Fiber Intake Quintile (g/day)**	**Q1** **12.94**	**Q2** **12.94–15.87**	**Q3** **15.90–19.00**	**Q4** **19.05–22.91**	**Q5** **>22.91**
Mean fiber intake (g)	10.5	14.6	17.4	20.8	27.6
Age	57.65	60.72	60.58	62.62	63.76
Females (%)	16 (17.4)	18 (19.4)	26 (28.0	27 (29.0)	30 (32.3)
Some college or more (%)	51 (56.0)	55 (59.1)	55 (60.4)	66 (71.0)	73 (79.3)
Stages III, IV (%)	65 (70.9)	66 (71.0)	66 (71.0)	63 (67.7)	58 (63.0)
Current smoker (%)	49 (53.3)	41 (44.1)	38 (40.9)	26 (28.3)	14 (15.2)
Current drinker (%)	72 (78.3)	61 (65.6)	69 (74.2)	55 (59.8)	62 (67.4)
Body Mass Index (kg/m^2^)	26.5	28.5	26.8	28.5	27.7
Total caloric intake (kcal)	1926.9	1971.3	1936.6	1947.4	1940.9
Glycemic load	116.1	118.4	119.6	127.9	125.0
Fruit/vegetable consumption (servings/day)	1.7	2.6	3.2	4.3	6.3
Total fat consumption (g)	68.8	76.9	76.1	71.0	67.4
**Whole grain intake quintile (g/day)**	**1** **13.70**	**2** **13.71–23.41**	**3** **23.43–32.95**	**4** **32.96–44.29**	**5** **>44.29**
Mean whole grain intake (g)	8.5	18.5	27.9	38.2	61.1
Age	60.15	62.44	59.02	61.26	62.31
Females (%)	18 (19.6)	28 (30.4)	21 (22.6)	25 (27.2)	24 (26.1)
Some college or more (%)	43 (47.3)	55 (60.4)	62 (67.4)	66 (71.7)	72 (78.3)
Stages III, IV (%)	58 (63.0)	64 (69.6)	71 (76.3)	67 (72.8)	81 (88.0)
Current smoker (%)	48 (52.2)	37 (40.2)	32 (34.4)	29 (31.9)	22 (23.9)
Current drinker (%)	66 (71.7)	60 (65.2)	70 (75.3)	65 (71.4)	58 (63.0)
Body Mass Index (kg/m^2^)	26.6	27.1	28.0	28.3	28.0
Total caloric intake (kcal)	2005.0	1826.3	2011.1	2022.3	1884.7
Glycemic load	114.1	120.0	120.9	123.1	132.6
Fruit/vegetable consumption (servings/day)	2.7	3.4	3.8	4.1	4.3
Total fat consumption (g)	74.0	74.9	71.0	72.5	68.4

**Table 3 nutrients-11-02304-t003:** Multivariable hazard ratios and their 95% confidence intervals of mortality risk by pretreatment total dietary fiber or whole grain consumption quintiles.

**Fiber Intake Quintile and Range (g/day)**	**1** **12.94**	**2** **12.94–15.87**	**3** **15.90–19.00**	**4** **19.05–22.91**	**5** **>22.91**	***p*_trend_^4^**
All-cause mortality						
^1^ Model 1	Referent	0.65 (0.39–1.10)	0.52 (0.30–0.90) *	0.66 (0.39–1.12)	0.34 (0.18–0.63) ***	0.002 **
^2^ Model 2	Referent	0.79 (0.46–1.35)	0.59 (0.34–1.05)	0.76 (0.45–1.31)	0.41 (0.21–0.78) **	0.014 *
^3^ Model 3	Referent	0.83 (0.43–1.59)	0.63 (0.32–1.25)	0.68 (0.30–1.52)	0.37 (0.14–0.95) *	0.04 *
**Fiber intake quintile and range (g/day)**	**1** **13.21**	**2** **13.41–16.03**	**3** **16.04–19.11**	**4** **19.22–23.17**	**5** **>23.20**	***p*_trend_^4^**
Cancer-specific mortality						
^1^ Model 1	Referent	0.84 (0.43–1.66)	0.73 (0.37–1.46)	0.76 (0.38–1.52)	0.48 (0.22–1.03)	0.06
^2^ Model 2	Referent	1.01 (0.51–2.01)	0.79 (0.39–1.63)	0.83 (0.41–1.69)	0.63 (0.28–1.40)	0.22
^3^ Model 3	Referent	1.10 (0.48–2.51)	0.80 (0.33–1.94)	0.68 (0.24–1.93)	0.46 (0.14–1.52)	0.14
**Whole grain intake quintile and range (g/day)**	**1** **13.70**	**2** **13.71–23.41**	**3** **23.43–32.95**	**4** **32.96–44.29**	**5** **>44.29**	***p*_trend_^4^**
All-cause mortality						
^1^ Model 1	Referent	0.96 (0.58–1.60)	0.55 (0.30–1.00) *	0.66 (0.37–1.15)	0.65 (0.38–1.13)	0.07
^2^ Model 2	Referent	0.88 (0.53–1.47)	0.60 (0.33–1.10)	0.71 (0.40–1.25)	0.65 (0.37–1.15)	0.12
^3^ Model 3	Referent	0.85 (0.50–1.46)	0.63 (0.33–1.20)	0.89 (0.47–1.68)	0.64 (0.34–1.24)	0.24
**Whole grain intake quintile and range (g/day)**	**1** **14.12**	**2** **14.23–23.97**	**3** **24.10–33.16**	**4** **33.16–44.42**	**5** **>44.60**	***p*_trend_^4^**
Cancer-specific mortality						
^1^ Model 1	Referent	1.16 (0.59–2.28)	0.84 (0.40–1.77)	0.91 (0.45–1.88)	0.80 (0.39–1.66)	0.08
^2^ Model 2	Referent	1.12 (0.57–2.20)	0.95 (0.45–2.03)	0.97 (0.47–2.00)	0.87 (0.41–1.87)	0.24
^3^ Model 3	Referent	1.17 (0.57–2.39)	0.92 (0.41–2.07)	1.22 (0.54–2.75)	0.83 (0.35–1.95)	0.18

* *p* < 0.05, ** *p* < 0.01, and *** *p* < 0.001.^1^ Basic model—controlled for sex and age. ^2^ Clinicopathological model—controlled for sex, age, HPV status, tumor stage, and tumor site. ^3^ Fully adjusted model—Multivariable Cox proportional hazards model fit with the following covariates: Sex, age, HPV status, tumor stage, tumor site, education status, mean fruit and vegetable consumption, glycemic load, total fat, BMI, smoking, and drinking status. ^4^
*p* value for a test of linear trend. Participant dietary fiber or whole grain intake level was set to the median of the subject’s respective quintile. This variable was subsequently modeled as a continuous term using Cox regression.

**Table 4 nutrients-11-02304-t004:** Propensity score-weighted multivariable hazard ratios and their 95% confidence intervals of all-cause mortality risk by pretreatment total dietary fiber consumption quintiles.

Fiber Intake Quintile and Range (g/day)	1 12.94	2 12.94–15.87	3 15.90–19.00	4 19.05–22.91	5 >22.91	*p* _trend_ ^2^
All-cause mortality						
^1^ Model 3	Referent	1.16 (0.72–1.85)	0.83 (0.52–1.33)	0.54 (0.31–0.95) *	0.22 (0.10–0.48) ***	<0.0001 ***

* *p* < 0.05 and *** *p* < 0.001. ^1^ Fully adjusted model—Multivariable Cox proportional hazards model fit with the following covariates: Sex, age, HPV status, tumor stage, tumor site, education status, mean fruit and vegetable consumption, glycemic load, total fat, BMI, smoking, and drinking status. ^2^
*p* value for a test of linear trend. Participant dietary fiber or whole grain intake level was set to the median of the subject’s respective quintile. This variable was subsequently modeled as a continuous term using Cox regression.

**Table 5 nutrients-11-02304-t005:** Multivariable hazard ratios and their 95% confidence intervals of recurrence risk by pretreatment total dietary fiber or whole grain consumption quintiles.

**Fiber Intake Quintile and Range (g/day)**	**1** **12.94**	**2** **12.94–15.87**	**3** **15.90–19.00**	**4** **19.05–22.91**	**5** **>22.91**	***p*_trend_^4^**
Recurrence						
^1^ Model 1	Referent	1.07 (0.60–1.88)	0.87 (0.48–1.57)	0.74 (0.40–1.36)	0.69 (0.37–1.28)	0.10
^2^ Model 2	Referent	1.31 (0.74–2.34)	0.97 (0.53–1.77)	0.85 (0.45–1.58)	0.93 (0.49–1.78)	0.43
^3^ Model 3	Referent	1.42 (0.73–2.75)	0.98 (0.49–1.98)	0.74 (0.32–1.73)	0.77 (0.30–1.97)	0.33
**Whole grain intake quintile and range (g/day)**	**1** **13.70**	**2** **13.71–23.41**	**3** **23.43–32.95**	**4** **32.96-44.29**	**5** **>44.29**	***p*_trend_^4^**
Recurrence						
^1^ Model 1	Referent	1.00 (0.58–1.75)	0.70 (0.38–1.27)	0.70 (0.38–1.27)	0.72 (0.40–1.30)	0.22
^2^ Model 2	Referent	0.96 (0.55–1.67)	0.72 (0.39–1.33)	0.87 (0.48–1.56)	0.81 (0.44–1.48)	0.53
^3^ Model 3	Referent	1.06 (0.59–1.92)	0.77 (0.40–1.50)	1.06 (0.56–2.04)	0.76 (0.38–1.50)	0.42

^1^ Basic model—Multivariable Cox proportional hazards model fit with the following covariates: Sex and age. ^2^ Multivariable model—Multivariable Cox proportional hazards model fit with the following covariates: Sex, age, HPV status, tumor stage, and tumor site. ^3^ Fully adjusted model—Multivariable Cox proportional hazards model fit with the following covariates: Sex, age, HPV status, tumor stage, tumor site, education status, mean fruit and vegetable consumption, glycemic load, total fat, BMI, smoking, and drinking status. ^4^
*p* value for a test of linear trend. Participant dietary fiber or whole grain intake level was set to the median of the subject’s respective quintile. This variable was subsequently modeled as a continuous term using Cox regression.

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
