# Peer review of "Dietary Fiber, Whole Grains, and Head and Neck Cancer Prognosis: Findings from a Prospective Cohort Study"

_nutrients, 2019, doi:10.3390/nu11102304_

Round 1

Reviewer 1 Report

 This manuscript reported that newly diagnosed head and neck patients who had higher dietary fiber intake, prior to the initiation of treatment, had better survival time. Although this is a very interesting finding, I did not see any evidence to support their conclusion that higher dietary fiber intake prolonged their survival time. There findings just showed that head and neck cancers arising in the patients who had higher dietary fiber intake might be biologically favourable.The authors should address this limitation in the Discussion.

Reviewer 2 Report

I would appreciate for reviewing this very interesting study that investigated the association between the intake of dietary fiber and whole grain and clinical outcomes in patients with HNSCC. Authors concluded that pretreatment higher dietary fiber intake may improve overall survival, but not disease-specific survival. In this cohort, there were 121 deaths and 77 disease-related deaths.

1, I would like to know what other cause of death (other than HNSCC) is. For example, cardiac and respiratory comorbidities are common among head and neck cancer survivors.

2, Because authors included only smoking and drinking status (never, former, current) in Cox proportional hazards models, residual confounding might be influenced on these results. Frequency or cumulative dose of cigarette smoking or alcohol drinking should be considered.

3, Table 1: As for Sex and Stage, total number of participants might be incorrect. (Sex= 461, Stage= 483)

4, As authors mentioned, it is big limitation that fiber consumption may decrease during and after cancer treatment in some patients.

5, Simply, higher fiber intake may be surrogated with good life style. 
